# META-LEARNING TO GUIDE SEGMENTATION

## ABSTRACT

There are myriad kinds of segmentation, and ultimately the "right" segmentation of a given scene is in the eye of the annotator. Standard approaches require large amounts of labeled data to learn just one particular kind of segmentation. As a first step towards relieving this annotation burden, we propose the problem of guided segmentation: given varying amounts of pixel-wise labels, segment unannotated pixels by propagating supervision locally (within an image) and non-locally (across images). We propose guided networks, which extract a latent task representation—*guidance*—from variable amounts and classes (categories, instances, etc.) of pixel supervision and optimize our architecture end-to-end for fast, accurate, and data-efficient segmentation by meta-learning. To span the few-shot and many-shot learning regimes, we examine guidance from as little as one pixel per concept to as much as 1000+ images, and compare to full gradient optimization at both extremes. To explore generalization, we analyze guidance as a bridge between different levels of supervision to segment classes as the union of instances. Our segmentor concentrates different amounts of supervision of different types of classes into an efficient latent representation, non-locally propagates this supervision across images, and can be updated quickly and cumulatively when given more supervision.

## 1 INTRODUCTION

Many tasks of scientific and practical interest require grouping pixels, such as cellular microscopy, medical imaging, and graphic design. Furthermore, a single image might need to be segmented in several ways, for instance to first segment all people, then focus on a single person, and finally pick out their face. Learning a particular type of segmentation, or even extending an existing model to a new task like a new semantic class, generally requires collecting and annotating a large amount of data and (re-)training a large model for many iterations. Interactive segmentation with a supervisor in-the-loop can cope with less supervision, but requires at least a little annotation for each image, entailing significant effort over image collections or videos. Faced with endless varieties of segmentation and countless images, yet only so much expertise and time, a segmentor should be able to learn from varying amounts of supervision and propagate that supervision to unlabeled pixels and images.

We frame these needs as the problem of *guided* segmentation: given supervision from few or many images and pixels, collect and propagate this supervision to segment any given images, and do so quickly and with generality across tasks. The amount of supervision may vary widely, from a lone annotated pixel, millions of pixels in a fully annotated image, or even more across a collection of images as in conventional supervised learning for segmentation. The number of classes to be segmented may also vary depending on the task, such as when segmenting categories like cats vs. dogs, or when segmenting instances to group individual people. Guided segmentation extends few-shot learning to the structured output setting, and the non-episodic accumulation of supervision as data is progressively annotated. Guided segmentation broadens the scope of interactive segmentation by integrating supervision across images and segmenting unannotated images.

As a first step towards solving this novel problem, we propose guided networks to extract *guidance*, a latent task representation, from variable amounts of supervision (see Figure 1). To do so we meta-learn how to extract and follow guidance by training episodically on tasks synthesized from a large, fully annotated dataset. Once trained, our model can quickly and cumulatively incorporate annotations to perform new tasks not seen during training. Guided networks reconcile static and interactive modes of inference: a guided model is both able to make predictions on its own, like a fully supervised model, and to incorporate expert supervision for defining new tasks or correcting errors,

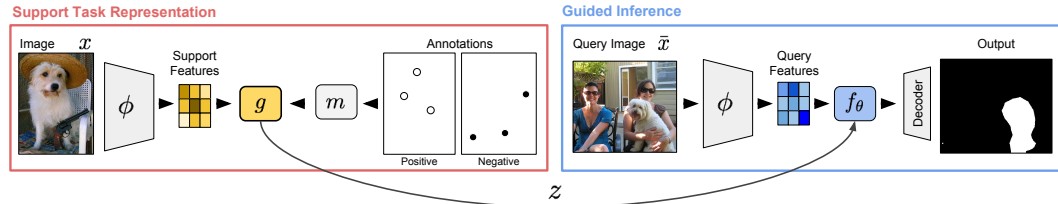

Figure 1: A guide $g$ extracts a latent task representation $z$ from an annotated image (red) for inference by $f_\theta(\bar{x}, z)$ on a different, unannotated image (blue).

like an interactive model. Guidance, unlike static model parameters, does not require optimization to update: it can be quickly extended or corrected during inference. Unlike annotations, guidance is latent and low-dimensional: it can be collected and propagated across images and episodes for inference without the supervisor in-the-loop as needed by interactive models.

We evaluate our method on a variety of challenging segmentation problems in Section 5: interactive image segmentation, semantic segmentation, video object segmentation, and real-time interactive video segmentation, as shown in 2. We further perform novel exploratory experiments aimed at understanding the characteristics and limits of guidance. We compare guidance with standard supervised learning across the few-shot and many-shot extremes of support size to identify the boundary between few-shot and many-shot learning for segmentation. We demonstrate that in some cases, our model can generalize to guide tasks at a different level of granularity, such as meta-learning from instance supervision and then guiding semantic segmentation of categories.

## 2 RELATED WORK

Guided segmentation extends few-shot learning to structured output models, statistically dependent data, and variable supervision in amount of annotation (shot) and numbers of classes (way). Guided segmentation spans different kinds of segmentation as special cases determined by the supervision that constitutes a task, such as a collection of category masks for semantic segmentation, sparse positive and negative pixels in an image for interactive segmentation, or a partial annotation of an object on the first frame of a clip for video object segmentation.

**Few-shot learning** Few-shot learning (Fei-Fei et al., 2006; Lake et al., 2015) holds the promise of data efficiency: in the extreme case, one-shot learning requires only a single annotation of a new concept. The present wave of methods (Koch et al., 2015; Santoro et al., 2016; Vinyals et al., 2016; Wang & Hebert, 2016; Bertinetto et al., 2016; Finn et al., 2017; Ravi & Larochelle, 2017; Snell et al., 2017) frame it as direct optimization for the few-shot setting: they synthesize episodes by sampling supports and queries, define a task loss, and learn a task model for inference of the queries given the support supervision. While these works address a setting with a fixed, small number of examples and classes at meta-test time, we explore settings where the number of annotations and classes is flexible.

Our approach is most closely related to episodically optimized metric learning approaches. We design a novel, efficient segmentation architecture for metric learning, inspired by Siamese networks (Chopra et al., 2005; Hadsell et al., 2006) and few-shot metric methods (Koch et al., 2015; Vinyals et al., 2016; Snell et al., 2017) that learn a distance to retrieve support annotations for the query. In contrast to existing meta-learning schemes, we examine how a meta-learned model generalizes across task families with a nested structure, such as performing semantic segmentation after meta-learning on instance segmentation tasks.

**Segmentation** There are many kinds of segmentation, and many current directions for deep learning techniques (Garcia-Garcia et al., 2017). We take up semantic (Everingham et al., 2010; Liu et al., 2011), interactive (Kass et al., 1988; Boykov & Jolly, 2001), and semi-supervised video object segmentation (Pont-Tuset et al., 2017) as challenge problems for our unified view with guidance. See Fig. 2 for summaries of these tasks.

For semantic segmentation Shaban et al. (2017) proposes a one-shot segmentor (OSLSM), which requires few but densely annotated images, and must independently infer one annotation and class at a time. Our guided segmentor can segment from sparsely annotated pixels and perform multi-way inference. For video object segmentation one-shot video object segmentation (OSVOS) by Caelles

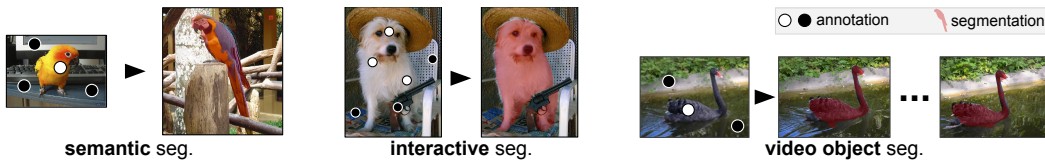

Figure 2: Guided segmentation groups different kinds of segmentation in one problem statement.

et al. (2017) achieve high accuracy by fine-tuning during inference, but this online optimization is too costly in time and fails with sparse annotations. Our guided segmentor is feed-forward, hence quick, and segments more accurately from extremely sparse annotations. Chen et al. (2018) impressively achieve state-of-the-art accuracy and real-time, interactive video object segmentation by replacing online optimization with offline metric learning and nearest neighbor inference on a deep, spatiotemporal embedding; however, they focus exclusively on video segmentation. We consider a variety of segmentation tasks, and investigate how guidance transfers across semantic and instance tasks and how it scales with more annotation. For interactive segmentation, Xu et al. (2016); Maninis et al. (2018) learn state-of-the-art interactive object segmentation, and Maninis et al. (2018) only needs four annotations per object. However, these purely interactive methods infer each task in isolation and cannot pool supervision across tasks and images without optimization, while our guided segmentor quickly propagates supervision non-locally between images.

## 3 GUIDED SEGMENTATION

Akin to few-shot learning, we divide the input into an annotated support, which supervises the task to be done, and an unannotated query on which to do the task. The common setting in which the support contains $K$ distinct classes and $S$ examples of each is referred to as $K$-way, $S$-shot learning (Lake et al., 2015; Fei-Fei et al., 2006; Vinyals et al., 2016). For guided segmentation tasks we add a further pixel dimension to this setting, as we must now consider the number of support pixel annotations for each image, as well as the number of annotated support images. We denote the number of annotated pixels per image as $P$, and consider the settings of $(S, P)$-shot learning for various $S$ and $P$. In particular, we focus on sparse annotations where $P$ is small, as these are more practical to collect, and merely require the annotator to point to the segment(s) of interest. This type of data collection is more efficient than collecting dense masks by at least an order of magnitude (Bearman et al., 2016). Since segmentation commonly has imbalanced classes and sparse annotations, we consider mixed-shot and semi-supervised supports where the shot varies by class and some points are unlabeled. This is in contrast to the standard few-shot assumption of class-balanced supports.

We define a guided segmentation task as the set of input-output pairs $(\mathcal{T}_i, \mathcal{Y}_i)$ sampled from a task distribution $\mathcal{P}$, adopting and extending the notation of Garcia & Bruna (2018). The task inputs are

$$\mathcal{T} = \left\{ \{(x_1, L_1), \ldots (x_S, L_S)\} \cup \{\bar{x}_1, \ldots, \bar{x}_Q\} \; ; \; x_s, \bar{x}_q \sim \mathcal{P}_l(\mathbb{R}^N) \right\}$$
$$L_s = \{(p_j, l_j) : j \in \{1 \ldots P\}, \; l \in \{1 \ldots K\} \cup \{\varnothing\}\}$$

where $S$ is the number of annotated support images $x_s$, $Q$ is the number of unannotated query images $\bar{x}_q$, and $L_s$ are the support annotations. The annotations are sets of point-label pairs $(p, l)$ with $|L_s| = P$ per image, where every label $l$ is one of the $K$ classes or unknown ($\varnothing$). The task outputs, that is the targets for the support-defined segmentation task on the queries, are

$$\mathcal{Y} = (y_1, \ldots, y_Q), \quad y_q = \{(p_j, l_j) : p_j \in \bar{x}_q\}$$

Our model handles general way $K$, but for exposition we focus on binary tasks with $K = 2$, or $L = (+, -)$. We let $Q = 1$ throughout as inference of each query is independent in our model.

## 4 GUIDED NETWORKS

Our approach to guided segmentation has two parts: (1) extracting a task representation from the semi-supervised, structured support and (2) segmenting the query given the task representation. We define the task representation as $z = g(x, +, -)$, and the query segmentation guided by that

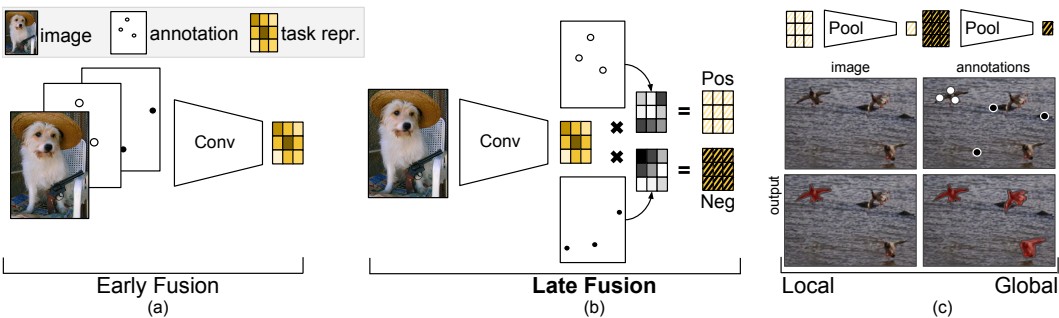

Figure 3: Extracting a task representation or "guidance" from the support. (a) Early fusion simply concatenates the image and annotations. (b) Our late fusion factorizes into image and annotation streams, improves accuracy, and updates quickly given new annotations. (c) Globalizing the task representation propagates appearance non-locally: a single bird is annotated in this example, but global guidance causes all the similar-looking birds to be segmented (red) regardless of location.

representation as $\hat{y} = f(\bar{x}, z)$. The design of the task representation $z$ and its encoder $g$ is crucial for guided segmentation to handle the hierarchical structure of images and pixels, the high and variable dimensions of images and their pixelwise annotations, the semi-supervised nature of support with many unannotated pixels, and skewed support distributions.

We examine how to best design the guide $g$ and inference $f$ as deep networks. Our method is one part architecture and one part optimization. For architecture, we define branched fully convolutional networks, with a guide branch for extracting the task representation from the support with a novel late fusion technique (Section 4.1), and an inference branch for segmenting queries given the guidance (Section 4.2). For optimization, we adapt episodic meta-learning to image-to-image learning for structured output (Section 4.3), and increase the diversity of episodes past existing practice by sampling within and *across* segmentation task families like categories and instances.

## 4.1 GUIDANCE: FROM SUPPORT TO LATENT TASK REPRESENTATION

The task representation $z$ must fuse the visual information from the image with the annotations in order to determine what should be segmented in the query. As images with (partial) segmentations, our support is statistically dependent because pixels are spatially correlated, semi-supervised because the full supervision is arduous to annotate, and high dimensional and class-skewed because scenes are sizable and complicated. For simplicity, we first consider a binary task with $(1, P)$-shot support consisting of one image with an arbitrary number of annotated pixels $P$, and then extend to multi-way tasks and general $(S, P)$-shot support. To begin we decompose the support encoder $g(x_s, +_s, -_s)$ across receptive fields indexed by $i$ for local task representations $z_i = g(x_{si}, +_{si}, -_{si})$; this is the same independence assumption made by existing fully convolutional approaches to structured output. See Figure 3 for an overview and our novel late global fusion technique.

**Early Fusion (prior work)** Stacking the image and annotations channel-wise at the input makes $z_{si} = g_{\text{early}}(x, +, -) = \phi_S(x \oplus + \oplus -)$ with a support feature extractor $\phi_S$. This early fusion strategy, employed by Xu et al. (2016), gives end-to-end learning full control of how to fuse. Masking the image by the positive pixels (Shaban et al., 2017; Yoon et al., 2017) instead forces invariance to context, potentially speeding up learning, but precludes learning from the background and disturbs input statistics. All early fusion techniques suffer from an inherent modeling issue: incompatibility of the support and query representations. Stacking requires distinct $\phi_S, \phi_Q$ while masking disturbs the input distribution. Early fusion is slow, since changes in annotations trigger a full pass through the network, and only one task can be inferred at a time, limiting existing interactive and few-shot segmentors alike (Xu et al., 2016; Maninis et al., 2018; Shaban et al., 2017).

**Late Fusion (ours)** We resolve the learning and inference issues of early fusion by factorizing features and annotations in the guide architecture as $z_{si} = g_{\text{late}}(x, +, -) = \psi(\phi(\bar{x}), m(+), m(-))$. We first extract visual features from the image alone by $\phi(x)$, map the annotations into masks in the feature layer coordinates $m(+), m(-)$, and then fuse both by $\psi$ chosen to be element-wise product. This factorization into visual and annotation branches defines the spatial relationship between image and annotations, improving learning sample efficiency and inference computation time. Fixing $m$

to interpolation and $\psi$ to multiplication, the task representation can be updated quickly by only recomputing the masking and not features $\phi$. See Figure 3 (center). We do not model a distribution over z, although this is a possible extension of our work for regularization or sampling diverse segmentations.

Our late fusion architecture can now share the feature extractor $\phi$ for joint optimization through the support and query. Sharing improves learning efficiency with convergence in fewer iterations and task accuracy with $60\%$ relative improvement for video object segmentation. Late fusion reduces inference time, as only the masking needs to be recomputed to incorporate new annotations, making it capable of real-time interactive video segmentation. Optimization-based methods (Caelles et al., 2017) need seconds or minutes to update.

**Locality** We are generally interested in segmentation tasks that are determined by visual characteristics and not absolute location in space or time, i.e. the task is to group pixels of an object and not pixels in the bottom-left of an image. When the support and query images differ, there is no known spatial correspondence, and the only mapping between support and query should be through features. To fit the architecture to this task structure, we merge the local task representations by $m_P(\{z_{si} : \forall i\})$ for all positions $i$. Choosing global pooling for $m_P$ globalizes the task by discarding the spatial dimensions. The pooling step can be done by averaging, our choice, or other reductions. The effect of pooling in an image with multiple visually similar objects is shown in Figure 3 (right).

**Multi-Shot and Multi-Way** The full $(S, P)$-shot setting requires summarizing the entire support with a variable number of images with varying amounts of pixelwise annotations. Note in this case that the annotations might be divided across the support, for instance one frame of a video may only have positives while a different frame has only negatives, so $S$-shot cannot always be reduced to 1-shot, as done in prior work Shaban et al. (2017). We form the full task representation $z_S = m_S(\{z_1, \ldots, z_S\})$ simply and differentiably by averaging the shot-wise representations $z_s$. While we have considered binary tasks thus far, we extend guidance to multi-way inference do in our experiments. We construct a separate guide for each class, averaging across all shots containing annotations for that class. Note that all the guides share $\phi$ for efficiency and differ only in the masking.

## 4.2 GUIDING INFERENCE

Inference in a static segmentation model is simply $\hat{y} = f_\theta(\bar{x})$ for output $y$, parameters $\theta$, and input $\bar{x}$. Guided inference is a function $\hat{y} = f(\bar{x}, z)$ of the query given the guidance extracted from the support. We further structure inference by $f(\phi(\bar{x}), z)$, where $\phi$ is a fully convolutional encoder from input pixels to visual features.

Multiple forms of conditioning are possible and have been explored for low-dimensional classification and regression problems by the few-shot learning literature. In preliminary experiments we consider parameter regression, nearest neighbor and prototype retrieval, and metric learning on fused features. We select metric learning with feature fusion because it was simple and robust to optimize. Note that feature fusion is similar to siamese architectures, but we directly optimize the classification loss rather than a contrastive loss.

In particular we fuse features by $m_f = \phi(x) \oplus \text{tile}(z)$ which concatenates the guide with the query features, while tiling $z$ to the spatial dimensions of the query. The fused query-support feature is then scored by a small convolutional network $f_\theta$ that can be interpreted as a learned distance metric for retrieval from support to query. For multi-way guidance, the fusions of the query and each guide are batched for parallel inference.

## 4.3 EPISODIC OPTIMIZATION AND TASK DISTRIBUTIONS

We distinguish between optimizing the parameters of the model during training (learning) and adapting the model during inference (guidance). Thus during training, we wish to "learn to guide." In standard supervised learning, the model parameters $\theta$ are optimized according to the loss between prediction $\hat{y} = f_\theta(x)$ and target $y$. We reduce the problem of learning to guide to supervised learning by jointly optimizing the parameters of the guidance branch $g$ and the segmentation branch $f$ according to the loss between $f_\theta(\bar{x}, z)$ and query target $y$, see Figure 4.

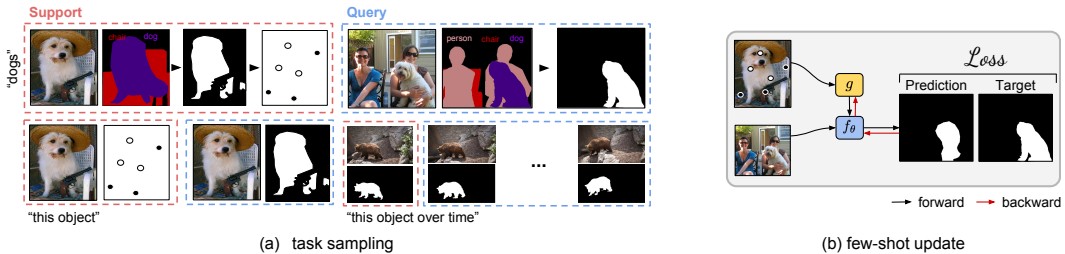

Figure 4: Optimization for guided segmentation. (a) Synthesizing tasks from densely annotated segmentation data. (b) One task update: episodic training reduces to supervised learning.

For clarity, we distinguish between tasks, a given support and query for segmentation, and task distributions that define a kind of segmentation problem. For example, semantic segmentation is a task distribution while segmenting birds (a semantic class) is a task. We train a guided network for each task distribution by optimizing episodically on sampled tasks. The supports and queries that comprise an episode are synthesized from a fully labeled dataset. We first sample a task, then a subset of images containing that task which we divide into support and query. During training, the target for the query image is available, while for testing it is not. We binarize support and query annotations to encode the task, and spatially sample support annotations for sparsity.

Given inputs and targets, we train the network with the pixelwise cross-entropy loss between the predicted and target segmentation of the query. See Sections 7.1 and 7.2 for more details on data processing and network optimization respectively.

After learning, the model parameters are fixed, and task inference is determined by guidance. While we evaluate for varying support size $S$, as described in 4.2, we train with $S = 1$ for efficiency while sampling $P \sim \text{Uniform}(1, 100)$. Once learned, our guided networks can operate at different $(S, P)$ shots to address sparse and dense pixelwise annotations with the same model, unlike existing methods that train for particular shot and way. In our experiments, we train with tasks sampled from a single task distribution, but co- or cross-supervision of distributions is possible. Intriguingly, we see some transfer between distributions when evaluating a guided network on a different distribution than it was trained on in Section 5.3.

## 5 RESULTS

We evaluate our guided segmentor on a variety of problems that are representative of segmentation as a whole: interactive segmentation, semantic segmentation, and video object segmentation. These are conventionally regarded as separate problems, but we demonstrate that each can be viewed as an instantiation of guided segmentation. As a further demonstration of our method, we present results for real-time, interactive video segmentation from dot annotations. To better understand the characteristics of guidance, we experiment with cross-task supervision in Section 5.2 and guiding with large-scale supports in Section 5.3.

To standardize evaluation we select one metric for all tasks: the intersection-over-union (IU) of the positives averaged across all tasks and masks. This choice allows us to compare scores across the different kinds of segmentation we consider without skew from varying numbers of classes or images. Note that this metric is not equivalent to the mean IU across classes that is commonly reported for semantic segmentation. Please refer to Section 7.3 for more detail.

We include fine-tuning and foreground-background segmentation as baselines for all problems. Fine-tuning simply attempts to optimize the model on the support. Foreground-background verifies that methods are learning to co-vary their output with the support supervision and sets an accuracy floor.

The backbone of our networks is VGG-16 (Simonyan & Zisserman, 2015), pre-trained on ILSVRC Russakovsky et al. (2015), and cast into fully convolutional form (Shelhamer et al., 2016). This choice is made for fair comparison with existing works across our challenge tasks of semantic, interactive, and video object segmentation without confounds of architecture, pre-training data, and so forth.

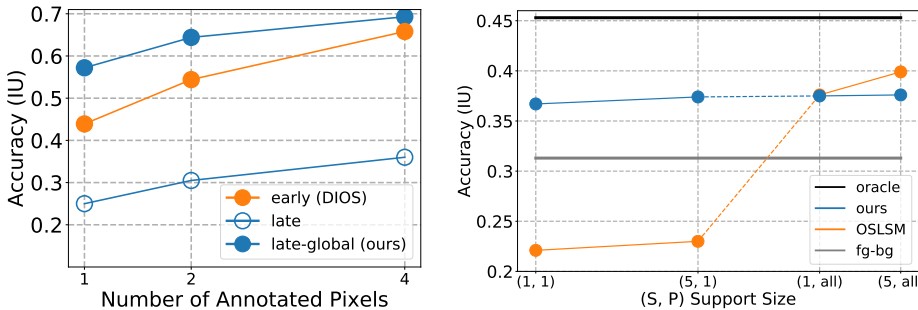

Figure 5: (left) Interactive segmentation of objects in images. (right) Guided semantic segmentation of held-out classes: we are state-of-the-art with only two points and competitive with full annotations.

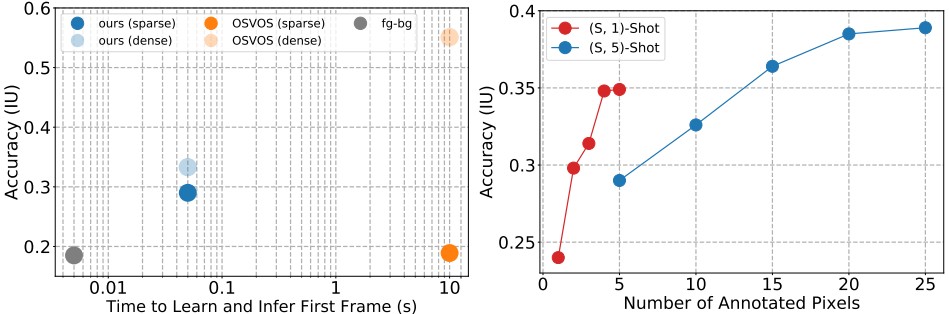

Figure 6: (left) Accuracy-time evaluation for sparse and dense video object segmentation on DAVIS'17 val. (right) Real-time interactive video segmentation on simulated dot interactions.

## 5.1 GUIDANCE FOR INTERACTIVE, VIDEO OBJECT, AND SEMANTIC SEGMENTATION

**Interactive Image Segmentation** We recover this problem as a special case of guided segmentation when the support and query images are identical. We evaluate on PASCAL VOC (Everingham et al., 2010) and compare to deep interactive object selection (DIOS) (Xu et al., 2016), because it is state-of-the-art and shares our focus on learning for label efficiency and generality. Our approach differs in support encoding: DIOS fuses early while we fuse late and globally. Our guided segmentor is more accurate with extreme sparsity and intrinsically faster to update, as DIOS must do a full forward pass. See Figure 5 (left). From this result we decide on late-global guidance throughout.

**Video Object Segmentation** We evaluate our guided segmentor on the DAVIS 2017 benchmark (Pont-Tuset et al., 2017) of 2–3 second videos. For this problem, the object indicated by the fully annotated first frame must be segmented across the video. We then extend the benchmark to sparse annotations to gauge how methods degrade. We compare to OSVOS (Caelles et al., 2017), a state-of-the-art online optimization method that fine-tunes on the annotated frame and then segments the video frame-by-frame. While Chen et al. (2018) presents impressive results on this task and on real-time interactive video segmentation without optimization, their scope is limited to video, and they employ orthogonal improvements that make comparison difficult. We were unable to reproduce their results in our own experimental framework. See Figure 6 (left) for a comparison of accuracy, speed, and annotation sparsity.

In the dense regime our method achieves 33.3% accuracy for 80% relative improvement over OSVOS in the same time envelope. Given (much) more time fine-tuning significantly improves in accuracy, but takes 10+min/video. Guidance is $\sim 200\times$ faster at 3sec/video. Our method handles extreme sparsity with little degradation, maintaining 87% of the dense accuracy with only 5 points for positive and negative. Fine-tuning struggles to optimize over so few annotations.

**Interactive Video Segmentation** By dividing guidance and inference, our guided segmentor can interactively segment video in real time. As an initial evaluation, we simulate interactions with randomly-sampled dot annotations. We define a benchmark by fixing the amount of annotation and measuring accuracy as the annotations are given. The accuracy-annotation tradeoff curve is plotted in

Figure 6 (right). Our guided segmentor improves with both dimensions of shot, whether images ($S$) or pixels ($P$). Our guided architecture is feedforward and fast, and faster still to update for changes to the annotations.

**Semantic Segmentation** Semantic segmentation is a challenge for learning from little data due to the high intra-class variance of appearance. For this problem it is crucial to evaluate on not only held-out inputs, but held-out classes, to be certain the guided learner has not covertly learned to be an unguided semantic segmentor. To do so we follow the experimental protocol of Shaban et al. (2017) and score by averaging across four class-wise splits of PASCAL VOC (Everingham et al., 2010), with has 21 classes (including background), and compare to OSLSM.

Our approach achieves state-of-the-art sparse results that rival the most accurate dense results with just two labeled pixels: see Figure 5 (right). OSLSM is incompatible with missing annotations, as it does early fusion by masking, and so is only defined for $\{0, 1\}$ annotations. To evaluate it we map all missing annotations to negative. Foreground-background is a strong baseline, and we were unable to improve on it with fine-tuning. The oracle is trained on all classes (nothing is held-out).

## 5.2 Guiding Classes from Instances

We carry out a novel examination of meta-learning with cross-task supervision. In the language of task distributions, the distribution of instance tasks for a given semantic category are nested in the distribution of tasks for that category. We investigate whether meta-training on the sub-tasks (instances) can address the super-tasks (classes). This tests whether guidance can capture an enumerative definition of a semantic class as the union of instances in that category.

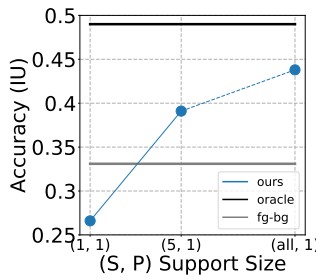

To do so, we meta-train our guided segmentor on interactive *instance* segmentation tasks draw from all classes of PASCAL VOC (Everingham et al., 2010), and then evaluate the model on *semantic* segmentation tasks from all categories. We experiment with $(S, 1)$ support from semantic annotations, where $S$ varies from one image to all the images in the training set, shown in the plot to the right. We compare to foreground-background as a class-agnostic accuracy floor, and a standard semantic segmentation net trained with semantic labels as an oracle. Increasing the amount of semantic annotations for guidance steadily increases accuracy.

## 5.3 Guiding by Few or Many Annotations

Thus far we have considered guidance in a variable but constrained scale of annotations, ranging from a single pixel in a single image to a few fully annotated images. We meta-learned our guided networks over episodes with such support sizes, and they perform accordingly well in this regime. Here we consider a much wider spectrum of support sizes, with the goal of understanding how guidance compares to standard supervised learning at both ends of the spectrum. To the best of our knowledge, this is the first evaluation of how few-shot learning scales to many-shot usage for structured output.

For this experiment we compare guidance and supervised learning on a transfer task between disjoint semantic categories. We take the classes of PASCAL VOC (Everingham et al., 2010) as source classes, and take the non-intersecting classes of COCO (Lin et al., 2014) as the target classes. We divide COCO 2017 validation into class-balanced train/test halves to look at transfer from a practical amount of annotation (thousands instead of more than a hundred thousand images). Our guided segmentor is meta-trained with semantic tasks sampled from the source classes, then guided with 5,989 densely annotated semantic masks from the target classes. For fair comparison, the supervised learner is first trained on the source classes, and then fine-tuned on the same annotated target data. Both methods share the same ILSVRC pre-training, backbone architecture, and (approximate) number of parameters. In this many-shot regime, guidance achieves 95% of supervised learning performance. A key point of this result is to shed light on the spectrum of supervision that spans few-shot and many-shot settings, and encourage future work to explore bridging the two.

## 6    Discussion

Guided segmentation unifies annotation-bound segmentation problems. Guided networks reconcile task-driven and interactive inference by extracting guidance, a latent task representation, from any amount of supervision given. With guidance our segmentor revolver can learn and infer tasks without optimization, improve its accuracy near-instantly with more supervision, and once-guided can segment new images without the supervisor in the loop.

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

# 7 APPENDIX

## 7.1 DATA PREPARATION

**Semantic Segmentation and Interactive Segmentation on PASCAL**  We use PASCAL VOC 2012 (Everingham et al., 2010) with the additional annotations of SBDD (Hariharan et al., 2011). We define the training set to be the union of the VOC and SBDD training sets, and take the validation set to be the union of VOC and SBDD validation sets, excluding the images in VOC val that overlap with SBDD train. We sparsify the dense masks with random sampling, which we found resulted in performance about equal to the more complex sampling strategies of Xu et al. (2016). Thus for a given $P$, we sample $P$ points randomly from each of the objects to be segmented, as well as the background. Labels for classes or instances that are not part of the task are relabeled to background. The process of sampling a task and sparsifying and remapping the ground truth labels is illustrated in Fig. 4.

For few-shot semantic segmentation, we follow the experimental protocol of Shaban et al. (2017). We test few-shot performance on held-out classes by dividing the 20 classes of PASCAL into 4 sets of 5 classes. Images that contain both held-out and training classes are placed in the held-out set. We subsample splits with more images to ensure that each split contains the same number of images. For

each split, we meta-train a guided segmentor with binary tasks sampled from the 15 training classes. We then compute the average performance across 1000 binary tasks sampled from the 5 held-out classes, and average across all four splits.

**Video Object Segmentation on DAVIS 2017** We use the DAVIS 2017 benchmark (Pont-Tuset et al., 2017) of 2-3s video clips. We meta-train on the training videos and report average performance on the validation videos. During training, we synthesize tasks by sampling any two frames from the same video and treating one as the support and the other as the query. During testing, the support consists of all labeled frames, while the remaining frames comprise the query. For the video object segmentation benchmark, the first frame is densely labeled. For interactive video segmentation, varying numbers of frames are labeled with varying numbers of pixelwise labels.

## 7.2 ARCHITECTURE AND OPTIMIZATION

The backbone of our guided networks as well as our baseline networks is VGG-16 (Simonyan & Zisserman, 2015), pre-trained on ILSVRC Russakovsky et al. (2015), and cast into fully convolutional form (Shelhamer et al., 2016). We largely follow the optimization procedure for FCNs detailed in (Shelhamer et al., 2016): we optimize our guided nets by SGD with a learning rate of $1e^{-5}$, batch size 1, high momentum $0.99$, and weight decay of $5e^{-4}$. The interpolation weights in the decoder are fixed to bilinear and not learned. Note that we normalize the loss by the number of pixels in each image in order to simplify learning rate selection across different datasets with varying image dimensions.

## 7.3 METRIC

Intersection-over-union (IU) is a standard metric for segmentation, but different families of segmentation tasks choose different forms of the metric. We report the IU of positives averaged across all tasks and masks, defined as $\frac{\sum_i tp_i}{\sum_i tp_i + fp_i + fn_i}$ where $i$ ranges over ground truth segment masks. This choice makes performance comparable across tasks, because it is independent of the number of classes. We choose not to include negatives in the metric because it adds no information, given the binary nature of the scoring, even for multi-class predictions and ground truth since these are handled as a set of binary tasks by the metric. Note that this metric is not directly comparable to the mean IU across classes typically reported for semantic segmentation benchmarks. As a point of comparison, the $0.62$ mean IU achieved by FCN-32s on the PASCAL segmentation benchmark corresponds to $0.45$ positive IU.

