# OpenReview forum: "Meta-Learning to Guide Segmentation"
_ICLR.cc/2019/Conference_

### Official Review · AnonReviewer2 · 2018-11-02
**Incremental idea and weak analysis**

**Rating:** 3
**Confidence:** 5

**Review:**

Summary
This paper proposes to formulate diverse segmentation problems as a guided segmentation, whose task is defined by the guiding annotations.
The main idea of this paper is using meta-learning to train a single neural network performing guidance segmentation.
Specifically, they encode S annotated support image into a task representation and use it to perform binary segmentation.
By performing episodic optimisation, the model's guidance to segmentation output is defined by the task distribution.

Strength
Learning a single segmentation algorithm to solve various segmentation problem is an interesting problem that worth exploring.
This paper tackles this problem and showed results on various segmentation problems.

Weakness
The proposed method, including the architecture and training strategy, is relatively simple and very closely related to existing approach. Especially, the only differences with the referenced paper (Shaban et al., 2017) is how the support is fused and how multiple guidance could be handled, which can be done by averaging. These differences are relatively minor, so I question the novelty of this paper.

This paper performs experiments on diverse tasks but the method is compared with relatively weak baselines absolute performance looks bad compared to existing algorithms exploiting prior knowledge for each of the tasks.
For example, the oracle performance in semantic segmentation (fully supervised method) is 0.45 IOU in PASCAL VOC dataset, while many existing algorithms could achieve more than 0.8 mean IOU in this dataset.
In addition, I question whether foreground / background baseline is reasonable baseline for all these tasks, because a little domain knowledge might already give very strong result on various segmentation tasks.
For example, in terms of video segmentation, one trivial baseline might include propagating ground truth labels in the first frame with color and spatial location similarity, which might be already stronger than the foreground / background baseline.

There are some strong arguments that require further justification.
- In 4.3, authors argue that the model is trained with S=1, but could operate with different (S, P).
However, it's suspicious whether this would be really true, because it requires generalisation to out-of-distribution examples, which is very difficult machine learning problem. The performance in Figure 5 (right) might support the difficulty of this generalisation, because increasing S does not necessarily increase the performance.
- In 5.3, this paper investigated whether the model trained with instances could be used for semantic segmentation. I think performing semantic segmentation with model trained for instance segmentation in the same dataset might show reasonable performance, but this might be just because there are many images with single instance in each image and because instance annotations in this dataset are based on semantic classes. So the argument that training with instance segmentation lead to semantic segmentation should be more carefully made.

Overall comment
I believe the method proposed in this paper is rather incremental and analysis is not supporting the main arguments of this paper and strength of the proposed method.
Especially, simple performance comparison with weak baselines give no clues about the property of the method and advantage of using this method compared to other existing approaches.

---

> ### Author Response · Authors · 2018-11-15
> **contributions, results metric, and interpretation of experiments**
>
> Thank you for the review and the attention to our architecture and results. Here we detail how our architectural choices lead to key differences from prior work, clarify the metric in our experiments, and discuss the interpretation of our experiments. We would appreciate if the reviewer can comment on how these points affect their views on the novelty, strength of results, and interpretation of our work and reconsider their rating.
>
> > the only differences with the referenced paper (Shaban et al., 2017) is how the support is fused and how multiple guidance could be handled, which can be done by averaging.
>
> Our work differs in architecture, optimization, and scope.
>
> For architecture, we factorize the approach into (1) extracting the task representation and (2) guiding inference by the representation: this takes the form of our novel late fusion architecture in contrast to the early fusion of Shaban et al. While this difference might appear minor, it has several important consequences. Late fusion allows for parameter sharing between guide and inference branches that makes optimization converge sooner. Given new support annotations, inference by our model updates an order of magnitude faster because only the late stage is recomputed, unlike the full recomputation of the net required by early fusion. For multi-class segmentation, our method only requires a single pass to compute a guide for each class, while Shaban et al. inefficiently require a forward pass per class since their early fusion is only defined for binary tasks.
>
> For optimization we meta-train on sparse annotations, not dense, and do not require per-branch learning rate tuning (since our parameters are shared). For tasks with sparsely labeled supports, we achieve an an accuracy improvement of ~50% relative over Shaban et al. for only two points per image; see Figure 5 (right).
>
> For scope, we formulate the more general problem of guided segmentation, and we agree with the reviewer that "learning a single segmentation algorithm to solve various segmentation problem is an interesting problem that worth exploring."  However Shaban et al. restrict their scope to one-shot semantic segmentation from densely labeled support. We hope that a unified meta-learning framework for varied types of segmentation leads to further progress on the accuracy of such methods over those that require more specialization.
>
> >  absolute performance looks bad compared to existing algorithms exploiting prior knowledge for each of the tasks
> > 0.45 IOU in PASCAL VOC dataset, while many existing algorithms could achieve more than 0.8 mean IOU in this dataset
>
> Please note that we score all methods with positive IU for consistency across tasks (section 5, paragraph 2), which is not equivalent to class-wise mean IU! We will further highlight and explain our choice of metric in the revision to resolve the commented on confusion, thank you.
>
> Our reported 0.45 positive IU oracle for few-shot semantic segmentation corresponds to 0.62 mean IU, which is expected for our FCN architecture based on VGG-16. The referred to methods that score more than 0.8 mean IU on PASCAL VOC require outside segmentation data, deeper architectures, longer optimization schedules, aggressive data augmentation, test-time post-processing, and more. These extensions are orthogonal to our scientific question comparing our general meta-learning method with the specialized methods for each of the tasks in a common experimental framework with the same base architecture.
>
> > I question whether foreground / background baseline is reasonable baseline for all these tasks
>
> The foreground-background baseline is surprisingly strong for video because DAVIS clips are biased towards containing one salient object per frame. To reduce the severity of this issue, our work evalutes on DAVIS'17 (Section 5.1, Figure 6) which includes some multi-object tasks instead of the simpler DAVIS'16.
>
> > In 4.3, authors argue that the model is trained with S=1, but could operate with different (S, P)
> > increasing S does not necessarily increase the performance.
>
> While the semantic-trained guided segmentor struggles to effectively aggregate larger supports, the instance-trained segmentor performs better with increasing S; see Section 5.2 for a discussion. Likewise our guided segmentor for video object segmentation improves with increasing S; see Section 5.1, Figure 6 (right).
>
> > In 5.3, this paper investigated whether the model trained with instances could be used for semantic segmentation.
> > but this might be just because there are many images with single instance in each image
>
> It’s true that PASCAL includes many images containing a single class. However, the semantic-guided instance-trained segmentor significantly outperforms the foreground-background baseline, which should do just as well on single-class images and single-instance images, so the accuracy of our guided segmentor cannot be entirely explained away by these kinds of images.

---

> > ### Comment · AnonReviewer2 · 2018-11-29
> > **Experimental Setting**
> >
> > Thank you for the detailed explanation about the experimental setting and clarification.
> > However, I'm still not convinced whether proposed model could learn diverse user's intent in case of interactive image segmentation.
> > I think there should be significant amount of ambiguity given few sparse guidance for segmentation.
> > For example, in case guidance is given on the chest of a person in a image, does it mean every person in the image should be segmented? only one person should be segmented? t-shirt should be segmented? or some region in a chest with similar pattern should be segmented.
> > In the experimental setting, it is hard to see how the proposed method is dealing with this ambiguity, and whether model's output is biased to class label in case guidance is clearly point some reason of image.
> > For example, guidance is pointing glasses on a face, does the model correctly segment glasses only? or does it segment whole face, or person.
> > I think these perspectives cannot be analyses just by comparing overall numbers with simple baselines.
> > I would recommend to present more analysis on the characteristics of the model by showing qualitative examples showing that the model correctly handle ambiguity and not bias.
> > One potential way to quantitatively evaluate this aspect would be using segmentation dataset containing classes with different granularity.
> > For example, one can repurpose MSCOCO dataset and include additional classes including subset of classes (e.g vegetable, food, fruit , etc) and designing an setting that ambiguity should be resolved.
> > I believe PASCAL VOC is somewhat limited to evaluate model's behaviour on such cases properly.
> >
> > I still believe this type of detailed analysis of the algorithm is essential to give acceptance to this paper.

---

### Official Review · AnonReviewer3 · 2018-11-03
**Unclear presentations, limited novelty.**

**Rating:** 3
**Confidence:** 4

**Review:**

Summary:
This paper proposed a few-shot learning approach for interactive segmentation. Given a set of user-annotated points, the proposed model learns to generate dense segmentation masks of objects. To incorporate the point-wise annotation, the guidance network is introduced. The proposed idea is applied to guided image segmentation, semantic segmentation, and video segmentation.

Clarity:
Overall, the presentation of the paper can be significantly improved. First of all, it is not clear what the problem setting of this paper is, as it seems to have two sets of training data of fully-annotated images (for training) and the combined set of point-wise annotated images and unannotated images (guidance images T in the first equation); It is not clear whether authors generate the second dataset out of the first one, or they have separate datasets for these two. Also, it is not clear how the authors incorporate the unannotated images for training.

The descriptions on model architecture are also not quite clear, as it involves two components (g and f) but start discussing with g without providing a clear overview of the combined model (I would suggest changing the order of Section 4.1 and Section 4.2 to make it clearer). The loss functions are introduced in the last part of the method, which makes it also very difficult to understand.

Originality and significance:
The technical contribution of the paper is very limited. I do not see many novel contributions in terms of both network architecture and learning perspective.

Experiment:
Overall, I am not quite convinced with the experiment results. The method is compared against only a few (not popular) interactive segmentation methods, although there exist many recent works addressing the same task (e.g. Xu et al. 2016).

The experiment settings are also not clearly presented. For instance, what is the dataset used for the evaluation of the first paragraph in section 5.1? How do you split the Pascal VOC data to exclusive sets? How do you sample point-wise annotation from dense mask labels? How does the sampling procedure affect the performance?

The performance of the guided semantic segmentation is also quite low, limiting the practical usefulness of the method. Finally, the paper does not present qualitative results, which are essential to understanding the performance of the segmentation system.

Minor comments:
1. There are a lot of grammar issues. Please revise your draft.
2. Please revise the notations in equations. For instance,
    T = {{(x_1, L_1),...} \cup {\bar{x}_1,...}
    L_s = {(p_j,l_j):j\in{1,...,P}, l\in{1,...,K}\cup{\emptyset}}
    Also, in the next equation, j\in\bar{x}_q} -> p_ j\in\bar{x}_q} (j is an index of pixel)

---

> ### Author Response · Authors · 2018-11-14
> **meta-learning setting, method novelty, and comparisons (1/2)**
>
> Thank you for your review, especially the comments regarding the clarity of the method description and experimental setting, which are helping us to revise the text to depend less on familiarity with meta-learning approaches and few-shot learning setups. In the meantime we offer clarifications here, and in particular address the problem setting, architecture and optimization novelty, and experimental comparisons. We will make a follow-up post once the revision is uploaded. Please let us know if the method and experiments are now clear, and how these details impact your evaluation of the submission's originality, significance, and experiments.
>
> > This paper proposed a few-shot learning approach for interactive segmentation
>
> We would like to clarify that our work is an extension of interactive segmentation. Our meta-learning learning approach, guided segmentation, generalizes the usual problem statement of interactive segmentation. Given an image with partial annotations, an interactive segmentor fully segments that image, but it cannot segment a new image without any annotations. That is, for an interactive segmentor, annotations on one image do not inform the segmentation of another image. On the other hand, our guided segmentor extracts a latent representation of the pixel-wise annotations and conditions on it to inform the segmentation of all images, and additional annotations on any image affect the segmentation of all of them.
>
> > I do not see many novel contributions in terms of both network architecture and learning perspective.
>
>
> Prior work is limited to binary segmentation of a single image (interactive segmentation by Xu et al. 2016), two-class tasks supervised by dense annotations from a single image (one-shot semantic segmentation by Shaban et al. 2017), and slow optimization that fails for sparse annotations (video object segmentation through fine-tuning by Caelles et al. 2017). Our novel choices for architecture and optimization are key to addressing these issues:
>
> - Our novel late-fusion architecture (Section 4.1 and Figure 3) is necessary for efficient representation and segmentation from annotations that are multi-shot (multi-image, multi-pixel) and multi-way (multi-class). Xu et al. and Shaban et al., with their early fusion architectures, are limited to one image and two classes at a time. When annotations change, they must re-compute the entire network as the annotations are fused early at the input, while we update in constant time w.r.t. the full network time since only the late stage is re-computed. For multi-class segmentation, our model simply and efficiently fuses shared image features with the annotations for each class (end of Section 4.1), while Xu et al. and Shaban et al. inefficiently have to do a forward pass for each class.
> - With optimization by meta-learning, our model learns to handle sparse annotations that the Caelles et al. approach of optimization by fine-tuning fails on. While Shaban et al. likewise optimize by meta-learning, they require dense annotations, and we show more than 50% relative improvement for accuracy in the sparse regime.
> - Our novel contributions to meta-learning optimization (Section 4.3) are (1) sampling tasks with different shot (number of labels) and way (number of classes) per episode of optimization for better generalization to different amounts of supervision and (2) investigating transfer learning when meta-learning one kind of task, instances, then meta-testing on a different kind of task, semantics.
>
> For novelty in experiments, our work is the first to show results on this set of tasks with a unified model.
>
> > The method is compared against only a few (not popular) interactive segmentation methods, although there exist many recent works addressing the same task (e.g. Xu et al. 2016)
>
> For comparison we chose popular, state-of-the-art at publication methods: DIOS (Xu et al. 2016) for interactive segmentation and OSVOS (Caelles et al.) for video object segmentation. To the best of our knowledge Shaban et al. 2017 is the first and only few-shot semantic segmentor prior to our work. Furthermore, these methods were chosen for fair comparison since their architectures and ours are all derived from a VGG-16 backbone and are free from confounding differences in post-processing, data augmentation, and so forth. Our work shows results on few-shot semantic segmentation, video object segmentation, and interactive instance segmentation (as mentioned above, guided segmentation is not simply interactive segmentation, as evidenced by this set of tasks).
>
> We ask that the reviewer please be specific about alternative comparisons.

---

> > ### Author Response · Authors · 2018-11-14
> > **meta-learning setting, method novelty, and comparisons (2/2)**
> >
> > > it is not clear what the problem setting of this paper is, as it seems to have two sets of training data of fully-annotated images (for training) and the combined set of point-wise annotated images and unannotated images (guidance images)
> >
> > Our problem setting is meta-learning for segmentation. Meta-learning seeks to learn a learning algorithm that can learn a new task, often from little supervision. In our case, a task consists of a support set of (sparsely) labeled images and a query set of unlabeled images to be segmented. In the standard terminology of few-shot learning, the "point-wise annotated images" are the labeled supports and the "unannotated images" are the queries to be segmented according to the labeled support.
> >
> > We divide the set of tasks into sets for meta-training and meta-testing. We optimize the parameters of our model to perform learning on tasks drawn from the meta-training set, and evaluate on tasks drawn from meta-test. For our guided nets, learning a task corresponds to inference in the model, which we call guidance: extracting the task representation from the supports and guided inference to segment the queries. Meta-training optimizes the model parameters to improve guidance, and once meta-training is complete the model parameters are fixed and only the task representation changes as a function of the support. For meta-testing we evaluate on heldout instances, classes, or videos in our interactive, semantic, and video object segmentation results respectively.
> >
> > > It is not clear whether authors generate the second dataset out of the first one, or they have separate datasets for these two.
> >
> > This kind of dataset division is a common approach to few-shot learning for image classification (e.g., Omniglot from Lake et al. 2015, miniImageNet from Vinyals et al. 2016) that we adapt to pixel-wise tasks.
> >
> > We generate our sparse meta-learning datasets from the standard, fully-annotated segmentation datasets by sampling different tasks (e.g., segment a particular bear in all the frames of the video) and subsampling the annotations. A task consists of a support set of (sparsely) labeled images and a query set of unlabeled images to be segmented. Tasks are synthesized from a densely labeled dataset such as PASCAL by binarizing and sparsifying dense masks, as illustrated in Section 4.3 Figure 4. During training, the query set is given as input to the model without labels, and the dense ground truth labels for the query set used to define the loss. We are revising section 4.3 to clearly explain this process.
> >
> > > it is not clear how the authors incorporate the unannotated images for training (guidance images)
> >
> > Our method is trained by meta-learning through episodic optimization: during meta-training, the unnannotated images are given as queries to be segmented by the model, the model infers an output segmentation, and these are compared against the true segmentation of the queries (known only during meta-training). Please see figure 4 and section 4.3. Are queries what was meant by guidance images?
> >
> > > what is the dataset used for the evaluation of the first paragraph in section 5.1? How do you split the Pascal VOC data to exclusive sets?
> >
> > The dataset used in the first paragraph of Section 5.1 is PASCAL VOC/SBD, as used in Xu et al., which we compare against (we are correcting this omission in a revision of the text—thank you for noticing it). For few-shot semantic segmentation, we follow the experimental protocol of Shaban et al., as stated in the second to last paragraph of Section 5.1, which tests few-shot performance on held-out classes by dividing the 20 classes of PASCAL into 4 sets of 5, then reports the average performance across these sets for the 5 held-out classes after training on the remaining 15. Images that contain both held-out and training classes are placed in the held-out set.
> >
> > > How do you sample point-wise annotation from dense mask labels? How does the sampling procedure affect the performance?
> >
> > The dense ground truth labels are sparsified via uniform random sampling. We found random sampling to perform about equal to more complex sampling strategies explored in previous work, such as Xu et al. 2016. We are adding these details to the paper appendix.
> >
> > > The performance of the guided semantic segmentation is also quite low
> >
> > The performance of our method is in some cases lower than the performance of task-specific methods (video object segmentation and 5-shot semantic segmentation). However, a main contribution of our work is to present a first general meta-learning framework for structured output tasks. A compensating advantage of our proposed late fusion architecture is that it is quicker to update than Shaban et al. and Caelles et al., making it more practical for interactive use.
> >
> > > Please revise the notations in equations.
> >
> > Thank you for noticing these typsetting errors! We are correcting them in a revision of the text.

---

### Official Review · AnonReviewer1 · 2018-11-03

**Rating:** 7
**Confidence:** 4

**Review:**

To my knowledge, this paper is probably the first one to apply few-shot learning concept into high-level computer vision tasks. In this paper's sense, segmentation. It proposes a general framework to few from the very few sample, extract a latent representation z, and apply it to do segmentation on a query. Cases of semantic, interactive and video segmentation are applied. Experiments are very thorough.

We see too many variants of few-shot learning papers on mini-imagenet or omniglot. For the reason of applying to high-level segmentation, the paper already deserves an acceptance for the first work. I believe this work would inspire many follow-ups in related domain (especially for high-level vision tasks)

Comments:

- what is interactive segmentation? I looked through the related work, it just mentioned some previous work without defining or describing it.

- z is the network output of g? is there any constraint on z? Like Gaussian distributions like what z is like in VAE models.

---

> ### Author Response · Authors · 2018-11-14
> **related work, interactive segmentation, and our latent task representation z**
>
> Thank you for the review and your enthusiasm for applying few-shot learning to richer visual tasks like segmentation! We provide a few clarifications and address the questions listed in your review. Given our response here, we would appreciate it if you could comment further regarding
>
> - novelty with respect to the one existing few-shot segmentation method we cite
> - clarity of our figure summarizing interactive segmentation and the other segmentation tasks we address (Figure 2)
>
> We agree that few-shot learning need not be limited to image classification and should address higher-level tasks such as different types of segmentation as we show in this work. We hope that our work inspires more progress on few-shot learning for structured output tasks for which labels are even more costly and scarce than image-level supervision.
>
> Our work is not the first to consider few-shot learning for structured output, but we do significantly generalize the problem scope and extend the approach. Shaban et al. (2017) consider one-shot semantic segmentation. We consider a wider range of tasks (instance, semantic, and video object segmentation), experiment with varying shot and way (from one-shot to 1000+ shot and 2-20 way) beyond the prior 1-5 shot and fixed 2-way of Shaban et al., and propose a novel late fusion architecture (that is faster to update during inference).
>
> > what is interactive segmentation?
>
> Interactive segmentation is the task of inferring dense segmentation masks from sparse pixel-wise labels within the same image (see middle panel of Figure 2 and our references Kass et al. 1998, Boykov and Jolly 2001, and Xu et al. 2016). Guided segmentation is our extension to interactive segmentation that can propagate pixel labels across images and not just within images. Guided segmentation is necessary to (1) cumulatively incorporate labels across inputs to keep improving the segmentation and (2) increase data efficiency by not requiring annotations on every input.
>
> > is there any constraint on z? Like Gaussian distributions like what z is like in VAE models
>
> z is the latent task encoding extracted by the guide branch g (see Figure 1 and Sections 4 & 4.1). We do not enforce a distribution over z, although this is a possible extension of our work for regularization or sampling diverse segmentations. We are revising the text to make it clear that there is no constraint on the value of z.

---

### Meta-Review · Area_Chair1 · 2018-12-17

**Confidence:** 4
**Recommendation:** Reject

**Metareview:**

Paper proposes a meta-learning approach to interactive segmentation. After the author response, R2 and R3 recommend rejecting this paper citing concerns of limited novelty and insufficient experimental evaluation (given the popularity of this topic in computer vision). R1 does not seem be familiar with the extensive literature on interactive segmentation and their positive recommendation has been discounted. The AC finds no basis for accepting this paper.